# Quinoxaline 1,4-Dioxides: Advances in Chemistry and Chemotherapeutic Drug Development

**DOI:** 10.3390/ph16081174

**Published:** 2023-08-17

**Authors:** Galina I. Buravchenko, Andrey E. Shchekotikhin

**Affiliations:** Gause Institute of New Antibiotics, 119021 Moscow, Russia; buravchenkogi@gmail.com

**Keywords:** heterocyclic *N*-oxides, quinoxaline 1,4-dioxides, methods of synthesis, biological and pharmaceutical activity, SAR-analysis, therapeutic application

## Abstract

*N*-Oxides of heterocyclic compounds are the focus of medical chemistry due to their diverse biological properties. The high reactivity and tendency to undergo various rearrangements have piqued the interest of synthetic chemists in heterocycles with *N*-oxide fragments. Quinoxaline 1,4-dioxides are an example of an important class of heterocyclic *N*-oxides, whose wide range of biological activity determines the prospects of their practical use in the development of drugs of various pharmaceutical groups. Derivatives from this series have found application in the clinic as antibacterial drugs and are used in agriculture. Quinoxaline 1,4-dioxides present a promising class for the development of new drugs targeting bacterial infections, oncological diseases, malaria, trypanosomiasis, leishmaniasis, and amoebiasis. The review considers the most important methods for the synthesis and key directions in the chemical modification of quinoxaline 1,4-dioxide derivatives, analyzes their biological properties, and evaluates the prospects for the practical application of the most interesting compounds.

## 1. Introduction

It is well known that nitrogen-containing heterocycles are privileged structures in medicinal chemistry because their derivatives have various biological properties [1]. The importance of nitrogen heterocycles as key pharmacophoric moieties lies in the synthesis of novel anticancer compounds [2,3,4,5,6]. Indeed, the presence of different nitrogen electron-donor atoms in the structure improves the interaction with target proteins, enzymes, and receptors through the formation of several types of interactions, such as hydrogen bonds, dipole-dipole, hydrophobic interactions, van der Waals forces, and π-stacking interactions. The chemistry of *N*-oxides of heterocyclic compounds is one of the dynamically developing areas of organic synthesis. Aliphatic and aromatic *N*-oxides are known as prodrugs due to their ability to be reduced by various oxidoreductases expressed in bacterial and tumor cells [7,8]. Quinoxaline 1,4-dioxides attract close attention due to their wide spectrum of biological activity. The presence of two *N*-oxide groups determines the pharmaceutical properties of quinoxaline-1,4-dioxides, such as antibacterial, antitumor, antifungal, insecticidal, herbicidal, and antiparasitic. The synthetic availability and biological potential of quinoxaline 1,4-dioxides define the prospects for their use in the targeted design of compounds with practical and valuable properties.

Moreover, the interest in heterocyclic *N*-oxides is explained by the presence of such structures in certain biologically active compounds found in natural sources. An example of a natural compound based on quinoxaline 1,4-dioxide is 6-chloro-2-quinoxalinecarboxylic acid 1,4-dioxide (**1**, Figure 1) produced by *Streptomyces ambofaciens*, which can inhibit the growth of gram-positive bacteria. It has been noted that esters and amides of this acid also have antibacterial activity [9]. Iodinin (**2**, Figure 1) is another example of a derivative of natural origin with a related structure to quinoxaline 1,4-dioxide [10]. Iodinin (**2**) was first identified and isolated from the bacterial culture *Chromobacterium iodinum* [11], and its antimicrobial properties were described in 1943 by H. McIlwain [12]. The results of recent studies have shown that iodinin (**2**) exhibits high cytotoxicity and selectivity for acute myeloid leukemia and promyelocytic leukemia cells (EC_50_ values for apoptotic cell death were 40 times lower than for non-tumor cells). Notably, iodinin (**2**) induced apoptosis in tumor cell isolates from patients with a poor survival prognosis [13,14].

Since the second half of the twentieth century, many researchers have observed the pronounced antibacterial activity of synthetic quinoxaline 1,4-dioxides. Some derivatives have been patented as antimicrobial agents and have found application in husbandry as promoters that increase animal weight gain [15].

Certain quinoxaline derivatives have also been patented and used in clinical practice as antibacterial drugs. For example, quinoxidine (**3**; 2,3-bis(acetoxymethyl)quinoxaline 1,4-dioxide, Figure 2) and dioxidine (**4**; 2,3-bis(hydroxymethyl)quinoxaline 1,4-dioxide, Figure 2) have been used in the clinic since the 1970s as broad-spectrum antibacterial agents [16].

Further studies have shown that quinoxaline 1,4-dioxides are a promising scaffold for the development of new drugs for the treatment of tuberculosis and parasitic infections such as malaria, trypanosomiasis, leishmaniasis, amoebiasis, and trichomoniasis [17]. For instance, derivative **5** (Figure 3) [18] has demonstrated promising antituberculosis activity, while compounds **6** and **7** exhibited high fungicidal [19] and antiviral [20] effects, respectively (Figure 3). Derivatives of quinoxaline-2-carbonitrile 1,4-dioxide, such as compound **8** (Figure 3), exhibit selective cytotoxicity against solid tumor cells under hypoxic conditions [21].

However, despite the high biological activity of quinoxaline 1,4-dioxides, the derivatives described in the literature have several disadvantages associated with low solubility, mutagenicity [22], photoallergic reactions [23,24], and the development of bacterial resistance [25].

Methods for the synthesis and biological activity of quinoxaline 1,4-dioxides have been previously discussed in several papers [26,27,28,29,30,31,32,33,34,35,36]. However, the reviews published in the last twenty years have primarily focused on the biological aspects of quinoxaline 1,4-dioxides, providing limited consideration of their chemical properties and structure-activity relationships. In addition, the current reviews on the chemistry and biology of quinoxaline 1,4-dioxides do not contain up-to-date information published since 2016. Moreover, the information regarding the synthesis methods and chemical modifications of this class of compounds has not been systematized yet. Therefore, there is an interest in the general analysis and summarization of the data in the field of synthesis, modification, and biological properties of quinoxaline 1,4-dioxides, as well as the coverage of emerging trends in their study in medicinal chemistry and pharmaceuticals.

## 2. Methods of the Synthesis of Quinoxaline 1,4-Dioxides

Synthetic approaches to the formation of quinoxaline 1,4-dioxides are significantly limited (Figure 4). Prior to 1965, the primary method for their synthesis was the direct oxidation of quinoxaline derivatives. Additionally, benzene-unsubstituted quinoxaline 1,4-dioxides can be obtained by the condensation of *o*-benzoquinone dioxime with 1,2-dicarbonyl derivatives. However, currently, a more efficient method for obtaining quinoxaline 1,4-dioxide derivatives has become the heterocyclization of benzofuroxans with enols or enamines by the Beirut reaction.

The oxidation of quinoxalines with peroxy acids or hydrogen peroxide does not result in a preparative yield of the desired quinoxaline 1,4-dioxides. This limitation arises from the deactivation of the heterocyclic core, preventing further electrophilic attack due to the formation of an *N*-oxide fragment following the oxidation of one of the nitrogen atoms [37]. In 2006, M. Carmeli and S. Rozen developed a method based on the application of a complex of hypofluorous acid with acetonitrile. This approach allows oxidizing of quinoxalines to quinoxaline 1,4-dioxides with quantitative yields, even for derivatives with electron-withdrawing substituents and sterically hindered 5-substituted derivatives. For instance, quinoxaline **9** can be oxidized to derivative **10** with a substituent at position 5, which is otherwise challenging to access using alternative methods (Figure 1) [37,38].

Several examples of the formation of a fragment of quinoxaline 1,4-dioxide via cyclization of *o*-benzoquinone dioxime with 1,2-dicarbonyl compounds are known. In 1970, E. Abushanab [39] described the interaction of 1,2-diketones with *o*-benzoquinone dioxime (**11**), leading to 2,3-disubstituted derivatives (compound **12**, Figure 2) in low yields. However, the formation of 2-hydroxyquinoxaline 1,4-dioxides as the main products, such as compound **13**, was observed in good yields when *α*-ketoaldehydes were used for this condensation (Figure 2) [39].

The most significant preparative method for the synthesis of quinoxaline 1,4-dioxides is the cyclization of benzofuroxans by the Beirut reaction, developed by M.J. Haddadin and C.H. Issidorides in 1965 [40].

Initially, quinoxaline 1,4-dioxides were obtained by the condensation of benzofuroxans with enamines. Thus, the interaction of benzofuroxan (**14a**) with morpholinylcyclohexene gives 2,3-tetramethylenequinoxaline 1,4-dioxide (**15**) in moderate yield (Figure 3) [41]. Later, the synthetic possibilities of the Beirut reaction were expanded by using a large number of starting substrates. In 1972, M.J. Haddadin and C.H. Issidorides used this method to synthesize phenazine 5,10-dioxides, analogs of the antibiotic Iodinin. The authors found that cyclization of benzofuroxan (**14a**) with hydroquinone in the presence of a base gives 2-hydroxyphenazine 5,10-dioxide (**16**) (Figure 3) [42]. In the same year, A. Tanaka et al. showed the possibility of using alkynes in the Beirut reaction for the synthesis of 2,3-disubstituted quinoxaline 1,4-dioxides. Thus, the derivative **17** was obtained from benzofuroxan (**14a**) and 2-(furan-2-ylethynyl)-5-nitrofuran in the presence of methylamine, although in low yield (Figure 3) [43]. Cyclization of benzofuroxan (**14a**) with aryl- and heteroaryl ethylenes (Figure 3) was proposed for the synthesis of 2-substituted quinoxaline 1,4-dioxides (for example, derivative **18**) by J. Li et al. [44].

Several possible variants of the Beirut reaction mechanism have been described [27,41,45,46]. The first stage of the proposed mechanism includes the nucleophilic attack of the formed enolate ion on the electrophilic nitrogen atom of benzofuroxan **14a**, leading to intermediate **a** [27]. Based on the mechanism of the Beirut reaction (Figure 5), the opening of benzofuroxan **14a** in one of the transition states is accompanied by an attack of the oxime nitrogen atom on the electrophilic carbonyl group of intermediate **b**, which causes the formation of dihydroxyquinoxaline **c**. Further elimination of water from intermediate **c** leads to the formation of the quinoxaline 1,4-dioxide (**d**).

In 1995, A. Monge et al. described the synthesis of 3-aminoquinoxaline-2-carbonitrile 1,4-dioxide **19a** by the Beirut reaction from malononitrile and benzofuroxan **14a** (Figure 4) when developing antiparasitic agents [47]. A similar cyclization with malonic ester in the presence of sodium hydride leading to 2-hydroxyquinoxaline 1,4-dioxide **20** in high yield was reported by Y. Xu et al. (Figure 4) [48].

Cyclization of benzofuroxans with 1,3-diketones, *β*-ketoesters, or *β*-ketoamides has been proposed to obtain 2-acyl derivatives of quinoxaline 1,4-dioxide [49]. Thus, the interaction of 5,6-difluorobenzofuroxan (**14b**) with acetylacetone, acetoacetic ester, or *N*-phenylacetoacetamide in triethylamine at 0–10 °C gives 6,7-difluoroquinoxaline 1,4-dioxides **21**–**22a**, **23**, respectively, with yields of 65–83% (Figure 5) [50].

T. Lima et al. evaluated the effect of acid-base catalysts in the Beirut reaction on the yield of 2-carboethoxyquinoxaline 1,4-dioxide during the study of the condensation of benzofuroxan (**14a**) with benzoylacetic esters [51]. Thus, the target products were obtained in the presence of organic bases in less than 7% yields. In the case of acid catalysis, the key derivatives formed slightly higher yields (~13%). Catalysis with potassium fluoride on alumina in the absence of an organic solvent leads to the formation of desired products in acceptable yields (31–40%). A higher yield (50–64%) of derivatives of 3-phenylquinoxaline-2-carboxylic acid 1,4-dioxide was achieved using K_2_CO_3_ as a catalyst in acetone or DMF.

T. Sun et al. carried out the heterocyclization of benzofuroxan **14a** and carbonyl compounds to substituted quinoxaline 1,4-dioxides via supramolecular catalysis by *β*-cyclodextrin in water [52]. A similar procedure was used to obtain conjugates of 6(7)-haloquinoxaline 1,4-dioxides with primaquine [53] or with *N*-aryl-substituted 1,2,3-triazoles [54], which have antimalarial and antimycobacterial activity.

In some cases, the Beirut reaction between monosubstituted benzofuroxans and CH-acids leads to a mixture of 7- and 6-substituted quinoxaline 1,4-dioxides [55,56,57], which is explained by the presence of tautomeric equilibrium in benzofuroxans [58,59]. A study of the regioselectivity of the Beirut reaction between monosubstituted benzofuroxans **14c**–**i** and benzoylacetonitrile revealed that, along with the 7-isomers **24a**–**g**, their 6-substituted analogs **25c**–**g** [60] are formed in the reaction (Figure 6). It is shown that the yield of the 6-isomer increases with the growth of electron-withdrawing properties of the substituent in the starting benzofuroxan and correlates well with the Hammett constant. Thus, the 6-isomer is not formed in the case of derivatives with electron-donating groups, while the 6-substituted compounds **25f**,**g** with CO_2_Me- and CF_3_-groups are the main products of the reaction. The structure of the isomers was accurately confirmed by NMR spectroscopy and X-ray analysis. Compounds **24a**–**g** and **25f**,**g** showed high antiproliferative activity against breast adenocarcinoma cells (MCF-7, MDA-MB-231) under hypoxic conditions. Moreover, the structural isomers **24c**–**g** and **25c**–**g** had considerable differences in properties.

The reaction of aminobenzofuroxans **14j**,**k** with benzoylacetonitrile in chloroform in the presence of Et_3_N also yields 7-amino-substituted quinoxaline-2-carbonitrile 1,4-dioxides **26a**,**b** (Figure 7) [61]. This method of heterocyclization was later used to synthesize 7-(piperazin-1-yl)-3-trifluoromethylquinoxaline 1,4-dioxide **27a** (Figure 7) [62]. The 7-amino derivatives of **26a**,**b** have high aqueous solubility and downregulate the expression of HIF1α, BCL2, and ERα and induce apoptosis in MCF-7 cells at submicromolar concentrations.

In summary, the Beirut cyclization is currently the main preparative method for the synthesis of quinoxaline 1,4-dioxides with various substituents at positions 2, 3, 6, and 7. Ongoing research is focused on studying regioselectivity and exploring the possibilities of varying starting compounds, catalysts, and conditions to increase the yield of target quinoxaline 1,4-dioxides.

## 3. Chemical Properties of Quinoxaline 1,4-Dioxides

The reactivity of quinoxaline 1,4-dioxides is generally similar to the properties of heteroaromatic *N*-oxides. As a result, most of the reactions of quinoxaline 1,4-dioxides described in the literature involve the modification of the *N*-oxide fragment or are related to the transformation of the functional groups of the side chains and the benzene ring of quinoxaline.

Since the *N*-oxide fragment in the quinoxaline 1,4-dioxide facilitates many reactions, its reduction holds important synthetic value for the preparation of quinoxaline derivatives. Deoxygenation of heteroaromatic *N*-oxides can proceed in the presence of various reducing agents, such as catalytic hydrogenation, complex metal hydrides, sodium hypophosphite, titanium (III) chloride, dissolving metals, ascorbic acid, trivalent phosphorus compounds, and a number of sulfur-containing compounds [63,64,65,66]. However, not all reagents are suitable for the preparative reduction of quinoxaline 1,4-dioxides to quinoxalines due to the tendency of these substrates to undergo deoxygenation to mono-*N*-oxides or the formation of dihydro- and tetrahydroquinoxalines [67].

Quinoxaline 1,4-dioxides are sensitive to UV irradiation and, as a result of photoinduced rearrangement, can yield various products that depend on the structure of the starting compound. In particular, 2-substituted quinoxaline 1,4-dioxides **28a**,**b** isomerize to 3-oxoquinoxaline 1-*N*-oxides **30a** and **30b** in high yield through the intermediate oxaziridines **29** under photolysis of their aqueous solutions (Figure 8) [68].

Reduction and rearrangements occurring in the 1,4-dioxide core are often observed under basic conditions [69,70,71,72,73], by treatment of nucleophiles [74,75,76] or acid halides [77,78,79], which frequently reduce the yield of the target products and limit the possibility of transforming the functional groups of quinoxaline 1,4-dioxides. At the same time, the electron-withdrawing nature of the *N*-oxide fragments in the quinoxaline nucleus makes it possible to carry out a nucleophilic substitution of the leaving groups in positions 2, 3 and 6, 7 of the heterocyclic core under relatively mild conditions.

### 3.1. Transformations of Alkyl and Acyl Groups

Transformations of alkyl groups in the quinoxaline core are one of the most important approaches for obtaining a wide range of derivatives. For example, the synthesis of the original antibacterial drugs quinoxidine and dioxidine (compounds **3** and **4**, Figure 2) is based on the modification of the methyl groups of the heterocycle. Bromination of 2,3-dimethylquinoxaline 1,4-dioxide (**12**) yields 2,3-bis(bromomethyl)quinoxaline 1,4-dioxide (**31**), which, upon treatment with acetic acid in the presence of triethylamine, gives quinoxidine (**3**). Hydrolysis of the di-*O*-acetyl derivative **3** in aqueous methanol leads to dioxidine (**4**) in good yield (Figure 9) [80].

The CH-groups in the methyl attached to the pyrazine ring of quinoxaline 1,4-dioxides have high acidity, which explains the facilitation of the condensation of 2-benzoyl-3-methylquinoxaline 1,4-dioxides **32a**,**b** with dimethylacetal of *N*,*N*-dimethylformamide (DMADMF) in *o*-xylene, which gives high yields of enamines **33a**,**b**, respectively (Figure 10) [81].

Similarly, enamine **34** was obtained from 2-carboethoxy-7-chloro-3-methylquinoxaline 1,4-dioxide and DMADMF [82]. The transamination of this derivative with morpholine gives enamine **35**, while the reaction with anilines is accompanied by cyclization to pyridin-2-ones, leading to the formation of derivatives **36a**–**c** in good yields (Figure 11).

The synthesis of aldehyde-derived quinoxaline 1,4-dioxides by the Beirut reaction is limited by the accessibility of the starting reagents. M. Quiliano et al. described the preparation of 2-formylquinoxaline 1,4-dioxides, developed for obtaining analogs of isoniazid [83,84]. Oxidation of 2,3-dimethylquinoxaline **12** by treatment with SeO_2_ under microwave irradiation gave 2-formylquinoxaline **37a**, which upon condensation with a derivative of hydrazine, resulted in hydrazones **38a**,**b** (Figure 12). Compounds **38a**,**b** showed high antitubercular activity against *M. tuberculosis* H37Rv with low cytotoxicity for non-tumor cells (Vero cells).

Another example of the use of 2-formylquinoxaline 1,4-dioxide for the synthesis of antimycobacterial agents was presented by Zhang H. et al. [85]. The condensation of carbaldehydes **39a**,**b** with amines formed the corresponding *N*-substituted 2-thiazolidinonequinoxaline 1,4-dioxides, which further underwent intramolecular cyclization with thioglycolic acid to give 2-thiazolidinone-substituted quinoxalines **40a**,**b** in good yield (Figure 13). Derivative **40b** demonstrated high antimycobacterial activity (MIC = 1.6 μg/mL, *M. tuberculosis* H37Rv), as well as significant antifungal activity (MIC = 2–4 μg/mL) against a panel of pathogenic fungi (*Candida* spp., *Aspergillus* spp., *Cryptococcus* spp.) comparable to the reference drug amphotericin B (MIC = 0.5–2 μg/mL).

In addition, 2-formylquinoxaline 1,4-dioxide **37a** was applied for the synthesis of nitrones **41a**,**b** via its condensation with *N*-substituted hydroxylamines (Figure 14) [86]. Derivatives **41a**,**b** exhibited high antibacterial activity against Gram-positive and Gram-negative bacteria, including resistant strains, both in vitro and in vivo. It has been shown that the introduction of an additional *N*-oxide fragment into position 3 leads to an increase in the selectivity of the antibacterial effect of these derivatives against Gram-negative bacteria.

An alternative synthesis of quinoxalinylnitrones based on the condensation of 2-methylquinoxaline 1,4-dioxide (**28b**) with *p*-nitrosodimethylaniline, resulted in a high yield of nitrone **42** (Figure 15) [87].

B.B. Michniak et al. described the conversion of 2-formylquinoxaline 1,4-dioxide (**39a**) to quinoxaline-2-carbonitrile 1,4-dioxide (**44**) via *O*-acyloxime **43** (Figure 16) [88].

Y. Pan et al. used the modification of the acetyl residue for the diversification of quinoxaline 1,4-dioxides and increasing the solubility of compounds in aqueous solutions [89]. Bromination of ketone **21b** and subsequent substitution of the halogen in *α*-bromoketone **45** with amines gave *α*-aminoketones **46a**,**b** in high yield (Figure 17). Treatment of the bromo derivative **45** with thiols yielded *β*-mercaptoketones **47a**,**b**. The obtained compounds demonstrated high antimycobacterial activity against the *M. tuberculosis* H37Rv strain without cytotoxic effects on non-tumor cells (Vero cells).

The condensation of 2-acetyl-3-methylquinoxaline 1,4-dioxide **21c** with *p*-chlorobenzaldehyde in NaOH solution leading to chalcone **48** with high yield (Figure 18) is described in several papers [81,90]. An example of the modification of chalcones based on quinoxaline 1,4-dioxide is the heterocyclization of compound **48** with thiourea and hydrazine, resulting in derivatives of thiopyrimidine **49** and pyrazole **50**, respectively (Figure 18) [81]. The obtained compounds **49**, **50** exhibit high antitumor activity against hepatocellular carcinoma HepG2 and are two orders of magnitude more active than the reference agent tirapazamine under hypoxic conditions.

An alternative method for the synthesis of chalcones based on quinoxaline 1,4-dioxide was developed by Burguete A. et al. [91]. The Wittig reaction of 2-acetyl-3-methylquinoxaline 1,4-dioxide **37b** under microwave irradiation yielded enone **51**, which has high antioxidant and anti-inflammatory activity (Figure 19) [92].

A. Monge’s group carried out cyclopropanation of the conjugated double bonds of chalcones based on quinoxaline 1,4-dioxide using the Corey-Tchaikovsky reaction [90]. Thus, the reaction of enone **51** with sulfoxonium methylide generated from trimethyloxosulfonium iodide (TMSOI) gave the cyclopropyl derivative **52** in low yield (Figure 20).

Compound **52** demonstrated good antimalarial activity against the *P. falciparum* FCR-3 strain.

### 3.2. Reactions of Amino Derivatives of Quinoxaline 1,4-Dioxides

A. Monge et al. demonstrated that 3-aminoquinoxaline-2-carbonitrile 1,4-dioxides **19b**–**d** can be acylated using acid chlorides, resulting in the formation of the corresponding amides **53a**–**d** (Figure 21) [93]. Condensation of 5-nitrofuran- and 5-nitrothiophene-2-carboxylic acids **53a**–**d** in the presence of carbodiimide (EDCI) and DMAP was used for the synthesis of the corresponding amides (Figure 21). Derivative **53a** exhibits MIC values close to the reference rifampicin against *M. tuberculosis* H37Rv. Compounds **53b** and **53c** have comparable activity with the anti-Chagas drug nifurtimox for the *T. cruzi* Tulahuen 2 strain.

The reaction of 2-aminoquinoxaline 1,4-dioxides **19a**,**b** with isocyanates leads to cyclic carbamates **54a**,**b**. The ring opening of these carbamates with alcohols results in the formation of alkylcarbamates of quinoxaline 1,4-dioxides **55a**,**b** (Figure 22) [94].

Soliman D.H. et al. described an interesting modification of the amino group in the pyrazine ring of quinoxaline, allowing the obtainment of tricyclic diazaphosphorins **57a**,**b** as a result of cyclization of 3-aminoquinoxaline 1,4-dioxides **19a** and **56** by treatment with phosphorus(V) sulfide in pyridine. Further methylation of diazaphosphorins **57a**,**b** yielded *N*,*S*,*S*-trimethyl derivatives **58a**,**b** (Figure 23). These compounds demonstrated high activity against the prostate cancer cells (PC3), comparable to doxorubicin [95]. In addition, compound **58b** was able to inhibit VEGFR-2 and SRC kinases (IC_50_ = 18 μM).

### 3.3. Metal-Containing Chelates of Quinoxaline 1,4-Dioxides

One of the promising strategies in the search for new antitumor agents is the development of metal-containing chelates [96]. For example, copper(II) complexes with amino acids [97], casiopeins [98], oligopeptides [99,100], and mono- and bis-thiosemicarbazones [101,102] have demonstrated significant antitumor activity in human tumor models in vivo. The study of complexes with transition metals showed that quinoxaline-2-carboxilic acid 1,4-dioxide could act as a mono- or bidentate ligand for different coordination ions (Cr(III), Mn(II), Fe(III), Co(II), Ni(II), Cu(II), Zn(II), Ce(III), Nd(III), V(IV)) [103].

Another example of a ligand for chelates is 3-aminoquinoxaline-2-carbonitrile 1,4-dioxides **19e**,**f**, which forms complexes with Cu(II) (**59**) [104] or V(IV) (**60**), respectively (Figure 24) [105].

Derivative **60** has caused hypoglycemic and antihyperlipidemic effects using in vitro models, while the copper complex **59** demonstrated promising antiproliferative potency and selectivity under hypoxic conditions.

### 3.4. Nucleophilic Substitution of Halogen Atoms

Despite the high biological activity of some quinoxaline 1,4-dioxide derivatives, their study is complicated by low solubility in pharmacologically acceptable aqueous media [106]. One way to increase the bioavailability of organic compounds is by introducing substituents containing salt-forming groups, such as amino groups. The presence of a di-*N*,*N*′-oxide fragment in the quinoxaline core increases the reactivity of quinoxalines in nucleophilic substitution reactions. Therefore, the most important methods of diversification of quinoxaline 1,4-dioxides are based on the modification of halogen atoms, alkoxy- or alkylthio-groups in the activated positions of the heterocycle.

M.A. Ortega et al. demonstrated that 3-aminoquinoxaline-2-carbonitrile 1,4-dioxides **19b** and **19e** do not undergo diazotization reactions under classical conditions due to the low basicity of the amino group and the low solubility of such compounds. However, it was possible to produce a diazonium salt and replace the diazo group with a halogen by the Sandmeyer reaction. Thus, treatment of amines **19b** and **19e** with *tert*-butyl nitrite, heating in acetonitrile in the presence of CuCl_2_ in an inert atmosphere gave 3-chloro derivatives **61a**,**b**, although in low yield (Figure 25) [107]. Further nucleophilic substitution of the chlorine atom in compounds **61a**,**b** by diamines in chloroform in the presence of K_2_CO_3_ led to 3-amino derivatives, such as piperazines **62a**,**b**, in low yields (Figure 25). An alternative method for introducing amine residues into position 3 of quinoxaline 1,4-dioxides is based on the substitution of alkylthio-, aryloxy-, and sulfonyl-groups, which gives 3-amino derivatives in high yields [108,109,110,111,112].

The substitution of halogen atoms in the benzene ring activated by electron-withdrawing groups in the pyrazine nucleus of quinoxalines proceeds more efficiently. Thus, a series of amino derivatives **64a**–**c** was synthesized by treating an excess of diamines with halogen derivatives of 3-phenylquinoxaline-2-carbonitrile 1,4-dioxides **24c** and **63a**,**b** in DMF, yielding moderate-to-good yields (40–85%) (Figure 26). It should be noted that for 6,7-dihalogen derivatives, the reaction proceeds regioselectively, leading to 6-amino-substituted **64b**,**c**, which is explained by the electron-withdrawing effect of the CN group in position 2 of the heterocycle [61,113]. The structure of these compounds was confirmed by the ^13^C NMR spectra, as well as X-ray analysis. Amino derivatives **64a**–**c** at low micromolar concentrations inhibited the proliferation of a panel of tumor cell lines of various tissue origins and also showed outstanding selectivity for malignant cells under hypoxia.

In 2021, a similar approach was used by N. Zhang et al. to modify 2-acyl-6,7-difluoro-3-methylquinoxaline 1,4-dioxides **21**–**22a**. The treatment of derivatives **21**–**22a** with piperazine or azoles in ethanol in the presence of K_2_CO_3_ yielded 6-amino derivatives **65a**–**c** (Figure 27), which exhibit promising antituberculosis potency [114].

A similar method for modifying 2-acyl-3-trifluoromethylquinoxaline 1,4-dioxides was used for the development of new antibacterial agents [62,115,116]. However, derivatives with two electron-withdrawing groups in positions 2 and 3 have a high tendency for deoxygenation of the 1,4-dioxide moiety [116,117]. The substitution of the halogen atom in derivatives **66a**–**c** proceeds more efficiently when using Boc-piperazine to give amino derivatives **67a**–**f** (Figure 28).

In the case of 6,7-dihalogen derivatives, the reaction also proceeds retrospectively but leads to 7-amino derivatives **67b**–**f**, the structure of which was confirmed by analysis of the ^13^C NMR spectra [62].

### 3.5. Transformations of Carboxylic Acids and Their Derivatives

1,4-Dioxides of quinoxaline-2-carboxylic acids and their derivatives have high biological activity and a wide range of pharmacological properties. Therefore, it is important to analyze their chemical properties and the most valuable methods of their transformations. The 1,4-dioxides of quinoxaline-2-carboxylic acids are unstable and easily decarboxylated. Thus, quinoxaline 1,4-dioxide **28a** is formed from acid **68** when heated in propanol (Figure 29) [56].

The hydrolysis of ester **22b** to acid **70** was successfully carried out by heating in aqueous ethanol in the presence of triethylamine and catalytic amounts of calcium chloride (Figure 30). It has been shown that calcium ions accelerate the hydrolysis of esters of 1,4-dioxides of quinoxaline-2-carboxylic acids due to the formation of complex **69** [118].

The direct alkaline hydrolysis of 2-carboethoxy-7-fluoroquinoxaline 1,4-dioxide **65b** proceeds in low yield and gives acid **71**, as described in [114]. This is apparently explained by the instability of quinoxaline 1,4-dioxides in the presence of bases (Figure 31).

S.S. Sabri et al. synthesized amides from quinoxaline-2-carboxylic acid **70** using diphenylphospharylazide (DPPA) as an activating agent. Carboxamides **72a**–**e** were obtained in good yields from acid **70** and primary amines in the presence of DPPA and triethylamine in DMF (Figure 32) [119].

The Curtius rearrangement can be used for the synthesis of some 2-amino derivatives of quinoxaline 1,4-dioxide from azides of 1,4-dioxides of quinoxaline-2-carboxylic acids. Hydrazinolysis and subsequent nitrosation of ester **22c** gave the azide of 3-methylquinoxaline-2-carboxylic acid **73**. Azide **73** rearranges into 2-alkyl carbamates **74a**–**c** or substituted ureas **75a**–**c** when boiled in an alcohol or amine solution, respectively (Figure 33) [120].

The transformations of the hydrazide of 3-methylquinoxaline-2-carboxylic acid **76**, obtained from the corresponding ethyl ester and hydrazine, are applied to the functionalization of the quinoxaline 1,4-dioxide scaffold at position 2 [82]. Condensation of hydrazide **76** with benzaldehydes in the presence of acetic acid in ethanol gave hydrazones **77a** and **77b** (Figure 34).

Treatment of hydrazide **76** with carbon disulfide leads to the cyclization of 2-(1,3,4-oxadiazol-2-thione) **78** (Figure 34). Boiling hydrazide **76** with *p*-nitrobenzoic acid in the presence of phosphoryl chloride gave 2-(1,3,4-oxadiazolyl)quinoxaline 1,4-dioxide **79**.

## 4. Chemotherapeutic Properties of Quinoxaline 1,4-Dioxides

The unique biological properties of quinoxaline 1,4-dioxides have attracted significant interest from scientists for their use as an important scaffold for drug development. As mentioned earlier, aliphatic and aromatic *N*-oxides are known as prodrugs due to their ability to be reduced by various oxidoreductases expressed in bacterial and tumor cells. Moreover, the attention given to heterocyclic *N*-oxides, particularly quinoxaline derivatives, can be explained by the presence of this scaffold in natural bioactive compounds, such as iodinin, its analog myxin, echinomycin, and aspergilic acid [121,122].

### 4.1. Antibacterial and Antifungal Activity of Quinoxaline 1,4-Dioxides

In 1938, the investigation of the biological properties of quinoxaline 1,4-dioxides started with the discovery of the antibacterial activity of its closest analog, iodinin [123,124]. Subsequent studies have revealed that the mechanism of the antibacterial action of quinoxaline 1,4-dioxides is associated with the ability of heterocyclic *N*-oxides to reduction by oxidoreductases, leading to the generation of ROS, DNA damage, inhibition of DNA and RNA synthesis, morphological changes in the bacterial cell wall, as well as suppression of extracellular bacterial nuclease and α-toxin action [125,126,127,128]. Thus, the single-electron reduction of NADPH-dependent monooxygenases (P450) of quinoxaline 1,4-dioxide (**28a**, Figure 8) leads to the formation of an oxygen-sensitive intermediate radical **80**, generating reactive hydroxyl radical and metabolite **81** (Figure 35) [129].

The medicinal drugs in this class of heterocycles include quinoxidine and dioxidine (compounds **3** and **4**, Figure 2), which were discovered in the 1970s and are still used in parenteral forms and in various topical drug formulations. Dioxidine (**4**, Figure 2) is a reserve drug for the treatment of purulent infections. Washing of burning and purulent-necrotic wounds with dioxidine promotes faster wound surface cleansing, stimulates wound regeneration, and enhances epithelization [130]. Quinoxidine (**3**, Figure 2) acts on both Gram-positive organisms (including *Staphylococcus* spp., resistant to other antimicrobial drugs) and Gram-negative bacteria (*K. pneumoniae*, *P. aeruginos*, *Proteus* spp.). Dioxidine (**4**, Figure 2) causes loosening of the cell wall and agglomeration of DNA fibrils in *Staphylococcus aureus* cells [113]. In anaerobic conditions, the antibacterial activity of this drug increases by 8–128 times, depending on the type of pathogen [113]. Dioxidine (**4**, Figure 2) does not exhibit cross-resistance with other antimicrobial drugs. Recent studies have confirmed the low toxicity of dioxidine, including the absence of cardiotoxicity, immunotoxicity, and toxic effects on the adrenal glands and other organs, which makes it suitable for use as a new dosage form for the treatment of pharyngitis and tonsillitis [131,132].

In the pursuit of quinoxaline 1,4-dioxides with antibacterial and antifungal activity, it has been shown that 2-phenylsulfonyl-substituted quinoxaline 1,4-dioxides hold promise for the development of new antimicrobial agents. Among the synthesized series, derivatives **82a**,**b** (Figure 6) demonstrated the highest activity against *Enterococcus faecalis* and *Enterococcus faecium* strains, with MIC values ranging from 0.4 to 1.9 μg/mL [133].

A. Carta et al. have discovered promising derivatives with high anti-candida activity against *Candida* spp. The authors have demonstrated that the 2-chlorinated derivatives **83a**,**b** exhibit the highest antifungal activity (MIC = 0.39–0.78 μg/mL, Figure 6) [134]. Furthermore, the water-soluble 3-trifluoromethyl derivatives **27a** and **67c** (Figure 6) have shown antimicrobial activity against Gram-positive and Gram-negative bacteria and mycobacteria, comparable to reference drugs ciprofloxacin and rifampicin, with MIC values ranging from 0.25 to 10 μg/mL [62]. Additionally, compounds **27a** and **67c** demonstrated high antifungal activity comparable to amphotericin B against fungal cultures of *C. albicans* ATCC 10,231 and *M. canis* B-200. An analysis of the structure-activity relationship of the 3-trifluoromethylquinoxaline 1,4-dioxide series (for example, derivatives **27a** and **67c**, Figure 6) revealed that the introduction of an additional piperazine residue negatively affects the antimicrobial properties of these compounds. Furthermore, substituents in position 2 of the quinoxaline core do not significantly influence the antimicrobial activity of such derivatives. In particular, 7-(piperazin-1-yl)-3-trifluoromethylquinoxaline 1,4-dioxides demonstrated high activity against Gram-positive bacteria, while their effect on Gram-negative bacteria and fungi was noticeably lower.

The study of the mechanism of antibacterial action on the dual reporter test system pDualrep2 revealed that the 3-trifluoromethyl derivatives **27a**, **67c** (Figure 6), as well as the reference dioxidine (**4**, Figure 2), induced a bacterial SOS-response in the strain *E. coli* BW25513 [62].

### 4.2. Antimycobacterial Activity of Quinoxaline 1,4-Dioxides

Previously, based on the structural similarity of quinoxaline 1,4-dioxides with vitamin K, which plays an important role in the metabolism of *M. tuberculosis* [135], A. Monge et al. tested a series of 3-aminoquinoxaline-2-carbonitrile 1,4-dioxides (**62a**, **84**, **85**, Figure 7, Figure 25) against clinical isolates of *M. tuberculosis* with MDR [136]. The screening results for antimycobacterial properties demonstrated high activity of some derivatives of this type (for example, compounds **62a**, **84**, **85**, Figure 7) against *M. tuberculosis* H37Rv strain, highlighting the significant role of *N*-oxide fragments in their ability to inhibit the growth of the mycobacterium [137,138,139]. Moreover, compound **62a** exhibited a high selectivity index (SI) (SI = IC_50,Vero_/MIC > 124).

The replacement of the nitrile group at position 2 of quinoxaline with a carboxamide moiety, acyl, or carboxyl groups increased the inhibitory potency and, more importantly, reduced the cytotoxicity of such derivatives by an order of magnitude (**21b**–**e**, **22b**–**d**, **86**–**87a**–**c**, Figure 8) [140,141]. Among the 2-carboxamide derivatives, compounds **62a** and **84**, **85** were the most active (Figure 8), with MIC values of 0.89, 0.42, and 0.14 μg/mL, respectively. In the series of 2-acyl derivatives, 2-acetylquinoxaline 1,4-dioxides **21b**–**e** showed the highest activity (Figure 8), although these derivatives had slightly higher MIC values (0.87, 3.13, 0.8, and 4.3 μg/mL, respectively) compared to their analogs with carboxamide (compounds **62a**, **84**, **85**) and ester (compounds **22b**–**d**, **87a**–**c**) groups. The quinoxaline-2-carboxylates **22b**–**d** and **87a**–**c** exhibited the highest activity against *M. tuberculosis* H37Rv among all tested derivatives (MIC = 0.56, 2.30, 1.50, and 0.01 μg/mL, respectively), surpassing the reference drug rifampicin in activity. Notably, compounds **87b** and **87c** showed activity against the majority of tested strains of *M. tuberculosis* with MDR.

Some studies have demonstrated that 3-alkylquinoxaline 1,4-dioxides with carboxamide (compounds **86a**–**c**), acyl (compounds **21b**–**e**), alkoxycarbonyl (compounds **22b**–**d**, **87a**–**c**), or thioaryl (compound **88a**) substituents at position 2 also possess high antituberculosis activity (Figure 8) [142]. Compound **88a** exhibited moderate MIC values (30–60 μg/mL); however, in bacteriostatic concentrations, it acted synergistically with isoniazid, reducing its MIC by more than two orders of magnitude [143]. 

Compounds **65b**–**d** (Figure 8) demonstrated high antimycobacterial activity against *M. tuberculosis* H37Rv, as well as low cytotoxicity for non-tumor cells (MIC ≤ 0.25 μg/mL, SI_(Vero)_ = 169–412) [114]. The 6-(1,3,4-triazol-1-yl) substituted derivative **65d** showed the highest antitubercular activity in a model of macrophages infected with *M. tuberculosis*. Additionally, compound **65d** disrupted the integrity of membranes and energy homeostasis of *M. tuberculosis* and significantly increased the level of ROS in eukaryotic cells. Consequently, it induced autophagy in infected macrophages, suggesting an indirect blockage of intracellular replication of *M. tuberculosis*. Pharmacological studies in vivo demonstrated that compound **65d** was reasonably safe and possessed favorable pharmacokinetic properties.

The analysis of the obtained data for 2-carbonyl-containing quinoxaline 1,4-dioxides (for example, compounds **21a**–**e**; **22b**–**d**; **87a**–**c** and **65b**–**d**, Figure 8) revealed that the acetyl group at position 2 of the quinoxaline led to moderate to good antimycobacterial activity. Replacement of this moiety with a butyryl or isovaleryl group dramatically reduced the activity. However, the ethoxycarbonyl group at the C_2_ carbon atom of the quinoxaline ring generally enhanced antimycobacterial activity. In contrast, a benzyl analog did not alter the activity of such derivatives. Nevertheless, the introduction of an isopropyl ester group at the 2 position leads to a significant decrease in antimycobacterial activity. Moreover, the presence of a piperidine, morpholine, or pyrrolidine group in the structure of derivatives containing a 2-carbonyl fragment negatively affected the activity against the *M. tuberculosis* strain. Interestingly, the introduction of an imidazole or a 1,2,4-triazole group at position 6 of the heterocyclic ring resulted in a significant increase in the anti-TB potency of the corresponding compounds (derivatives **65b**–**d**, Figure 8). This indicates that an aromatic nitrogen heterocyclic group attached to the C6 carbon atom may be beneficial for activity.

Compound **40b** (Figure 8) exhibited high antituberculosis activity (IC_50_ = 1.6 μg/mL) against *M. tuberculosis* H37Rv and showed high selectivity with weak cytotoxic effect (SI = 60.6) on non-tumor cells (Vero cells) [85].

### 4.3. Antiparasitic Activity of Quinoxaline 1,4-Dioxides

The resistance of parasites belonging to the *Plasmodium* genus, including *Plasmodium falciparum*, to antimalarial drugs such as chloroquine, mefloquine, and artemisin-based combination therapy, poses a significant challenge in the treatment and control of malaria [144]. A. Monge et al. have developed a series of quinoxaline 1,4-dioxides that exhibit activity against *P. falciparum*, both in vitro and in vivo [145]. Based on the SAR analysis, the authors have several derivatives with high activity against chloroquine-resistant *P. falciparum* FcB1 strain, including compounds **32c**, **62c**–**d**, **88b**–**d**, and **89a**,**b** (Figure 9) [146]. Among these, the 2-carbonitrile derivatives **89a** and **89b** (Figure 9) showed the highest activity, with an IC_50_ of 5–6 μM against *P. falciparum*. In an in vivo experiment using Rj:SWISS (IOPs Orl) mice infected with a resistant strain of *Plasmodium berghei* NK65 Vincke and Lips 1948 was shown that at four times administration of 20 mg/kg of derivatives **89a** and **89b** led to a suppression of parasitemia by 70% and 88% on the fifth day, respectively. However, increasing the dosage of the tested substances resulted in the toxic death of animals.

Mechanistic studies have shown that quinoxaline 1,4-dioxide derivatives act as irreversible inhibitors of peroxiredoxin-2 and exhibit strong lytic activity against *Plasmodium* spp. [147].

A. Monge et al. have recently disclosed the antileishmaniasis activity of 3-acylaminoquinoxaline-2-carbonitrile 1,4-dioxides (for example, derivatives **90**, **91**) [148,149]. Among tested compounds, **90**, **91** (Figure 10) exhibited the most activity against both amastigote and promastigote forms of *Leishmania amazonensis* while showing relatively low cytotoxicity for mammalian cells. In vitro and in silico studies revealed some structure-activity relationship correlations. First of all, the presence of a halogen atom at position 7 of quinoxaline was found to be critical for anti-leishmaniasis activity. Additionally, electron-donating substituents at position 7 were shown to reduce the potency of such derivatives by an order of magnitude. K.F. Chacón-Vargas et al. described a series of 7-isopropoxycarbonyl-substituted derivatives that inhibit the growth of amastigotes of *Leishmania mexicana*, with compounds **92**, **93** demonstrating the highest activity and selectivity against this parasite [150].

Further studies have revealed high activity against the causative agent of amoebiasis, Entamoeba histolytica, for certain quinoxaline 1,4-dioxides [151,152]. The A. Monge group designed and synthesized a library of 2-acyl-3-methylquinoxaline 1,4-dioxides. Notably, compounds containing the thienyl fragment at position 2, as well as electron-withdrawing groups in the benzene ring of quinoxaline, exhibited the highest antiamoebic potency [151]. The most active compound was 3-trifluoromethyl quinoxaline 1,4-dioxide **94** (Figure 11) with an IC_50_ value of 0.35 μM, although it showed relatively low selectivity for mammalian cells (SI [IC_50(Vero cells)_/IC_50(*E. histolytica*)_] = 16.7). The 3-methyl analog (compound **95**, Figure 11) was four times less active than the 3-trifluoromethyl derivative **94** (IC_50_ = 1.4 μM, *E. histolytica*), but it displayed a higher selectivity index (SI > 60).

A. Carta et al. also observed the antitrichomonas activity of 6,7-difluoro-3-methylquinoxaline 1,4-dioxides, including compounds **90b** and **96** (Figure 12) [142]. It was found that 2-phenylthio derivatives of 3-methylquinoxaline 1,4-dioxide exhibited 20–30 times higher activity than the reference drug metronidazole against the protist parasite *Trichomonas vaginalis* SS22. For instance, the derivative **90b** suppressed the growth of the protozoan at submicromolar concentrations (IC_50_ = 0.4 μM). Interestingly, sulfone **96** (Figure 12) [153] displayed four times lower activity (IC_50_ = 1.6 μM) than the non-oxidized analog **90b**, but it more effectively blocked the growth of *T. vaginalis* compared to the reference drug metronidazole (IC_50_ = 12.5 μM).

E. Torres and co-authors developed a series of quinoxaline 1,4-dioxides with activity against *Trypanosoma cruzi*, the causative agent of trypanosomiasis (Chagas disease), contained at position 2, the carbonitrile fragment, carboxyl or ester groups [154]. The evaluation of the biological properties of the synthesized compounds revealed promising derivatives (for example, compounds **53c**, **97**, Figure 13, [155]) with high activity against *T. cruzi*. It is shown that derivatives **53c**, **97** (Figure 13) exhibited IC_50_ values similar to the reference drug nifurtimox with a selectivity index SI > 10. The authors conducted a SAR analysis, revealing that electron-withdrawing substituents at position 3 and, more critically, at position 6 and/or 7 of quinoxaline play a key role in the inhibitory ability of these derivatives. Conversely, the introduction of electron-donating groups into the heterocyclic nucleus led to a decrease in the trypanosomicidal activity of quinoxaline 1,4-dioxides. The proposed mechanism of antitrypanosomal activity of the lead compounds **53c** and **97** (Figure 13) is based on the inhibition of mitochondrial dehydrogenases in *T. cruzi* cells.

Derivatives **67d**–**f** (Figure 13) also exhibited high antiparasitic activity against epimastigotes of *T. cruzi*, with IC_50_ < 1.5 μM, surpassing the activity of reference drugs nifurtimox and benznidazole [116,117].

The study of the relationship between structure and antiparasitic activity revealed the key role of the nitrile group at position 2 of quinoxaline 1,4-dioxide (for example, derivatives **53c**; **89a**,**b**; **62c**,**d**; **90**; **91**, Figure 9, Figure 10 and Figure 13). It should be noted that similar to the antitumor activity of such derivatives, replacing the carbonitrile group with carboxyl and carboxamide groups, which have close electronic influences, leads to a partial or complete loss of the activity of the compounds. It was found that the substituents at position 3 of the heterocycle also play an important role in the antiparasitic properties of quinoxaline 1,4-dioxides. In general, the presence of an aromatic fragment at the C_3_ carbon atom of quinoxaline 1,4-dioxides significantly increases the activity of such derivatives. Furthermore, electron-donating substituents at position 3 of the heterocycle positively affect the ability of the compounds to block the growth of protozoa and enhance the selectivity indexes of such derivatives against parasites. Additionally, the introduction of a halogen atom into the benzene ring of quinoxaline leads to an increased ability of the compounds to suppress the growth of parasites.

### 4.4. Anticancer Properties of Quinoxaline 1,4-Dioxides

It is known that quinoxaline-2-carbonitrile 1,4-dioxides have high antiproliferative activity against tumor cells of various histogenesis, and their activity increases in hypoxia. A voltammetric study of the electrochemical properties of series derivatives has shown that the introduction of electron-withdrawing groups into the benzene ring of quinoxaline is accompanied by a significant shift of the redox potential to the positive region. This shift correlates with an increase in the cytotoxicity of such compounds under hypoxia [156]. Taking into account the results of electrochemical studies, U. Das et al. synthesized 3-aminoquinoxaline-2-carbonitrile 1,4-dioxides **19a**–**f** (Figure 14), which selectively induced tumor cells death under hypoxic conditions with low cytotoxicity against non-malignant cells [157]. Further optimization of the structure of quinoxalines **19a**–**f** by introducing aminoalkylamino-groups into position 3 of quinoxaline (compounds **98a**–**d**, Figure 13) leads to an increase in the water solubility of such derivatives while retaining the high antitumor activity of the compounds [158]. It should also be noted that the coordination complexes of 3-aminoquinoxaline-2-carbonitrile 1,4-dioxides **59** and **60** with metals (Figure 24) exhibit high antiproliferative activity against solid tumor cells, with improved bioavailability and toxicity properties [159].

H.U. Gali-Muhtasib et al. [160] have shown that compounds **21f**, **99a**,**b** (Figure 15) induce apoptosis and inhibit the growth of colon carcinoma cells T-84 at micromolar concentrations, as well as inhibit the cell cycle in both normoxia and hypoxia. Compounds **21f**, **99a**,**b** were found to increase transcript levels of apoptotic transforming growth factor β1 (TGF-β1) and decrease the level of TGF-α mRNA. In addition, derivatives **21f**, **99a**,**b** suppress the expression of the antiapoptotic protein gene Bcl-2a and increase the expression level of proapoptotic proteins p53 and p21, which are key mediators that cause apoptosis and cell cycle blocking.

H.U. Gali-Muhtasib et al. observed the effect of quinoxaline 1,4-dioxides on the expression of mRNA transcription factor HIF-1*α* in T-cell leukemia [161]. It is known that the 1α subunit of the heterodimeric transcription factor HIF regulates a metabolic adaptation to oxygen deficiency and plays an important role in the transcriptional activation of genes involved in angiogenesis. The screening results showed that compounds **99a** and **99b** (Figure 15) effectively reduce the expression of HIF-1a mRNA, while their non-chlorinated analog **21f** does not affect the expression of the HIF-1*α* gene [162]. It was also found that compound **99a** induces the arrest of the cells in G2/M-phases and inhibits the expression of cyclin B, causing apoptosis of tumor cells. The derivative **99a** reduces the level of expression of the Bcl-2*α* gene encoding the antiapoptotic Bcl-2*α* protein and increases the expression of the proapoptotic BAX protein [163]. 

In 2008, Weng Q. et al. [164] found a highly active derivative **100** (Figure 16) capable of inhibiting the growth of a wide panel of tumor cell lines at low concentrations (IC_50_ = 0.2–8.9 µM) under hypoxic conditions. Recent studies, including virtual screening, the synthesis of polysubstituted derivatives, and the evaluation of their biological activity, have led to the discovery of a series of 3-phenylquinoxaline-2-carbonitrile 1,4-dioxides (compounds **26c**, **63a**, **64a** and **64d**, Figure 16) [61,114,165], which exhibit remarkable cytotoxicity. It was shown that compound **63a** (Figure 16) induces apoptosis in human tumor cells of various histogenesis. Moreover, lead compounds **26c** and **64a**,**d** (Figure 16) have in one order highest activity and selectivity under hypoxic conditions than the reference antitumor hypoxic cytotoxin tirapazamine [61,114].

Analysis of the structure-activity relationship in a series of antitumor agents based on quinoxaline 1,4-dioxide reveals an essential role for the carbonitrile group at position 2 of the heterocyclic ring (for example, derivatives **19a**–**f**; **26c**; **63a**; **64a**,**d**; **98a**–**d**, Figure 14 and Figure 16). Additionally, substituents at position 3 of the heterocycle are equally important for the antiproliferative properties of quinoxaline 1,4-dioxides. Among these compounds, congeners with a benzene ring at position 3 of the quinoxaline (such as compounds **21f**; **26c**; **63a**; **64a**,**d**; **99a**,**b**; **100**, Figure 15 and Figure 16) exhibit the highest activity, whereas compounds with different substituents in this position (for example, compounds **19a**–**f**; **98a**–**d**, Figure 14) show lower levels of activity. Another crucial factor for the ability of compounds to inhibit tumor cell growth under both normoxic and hypoxic conditions is the nature and location of substituents at positions 6 and 7 of quinoxaline 1,4-dioxide. Introduction of linear diamines into the benzene ring of quinoxaline results in a complete loss of antiproliferative activity, whereas the presence of cyclic diamines significantly increases activity, particularly under hypoxic conditions. Notably, derivatives containing substituents at position 7 of quinoxaline exhibit noticeably higher activity compared to their corresponding regioisomeric analogs with substituents at position 6. Furthermore, the introduction of a halogen atom into the quinoxaline structure positively affects the ability of compounds to inhibit tumor cell growth. Remarkably, the addition of a chlorine atom proves to be the most crucial modification for enhancing the hypoxic cytotoxicity of quinoxaline 1,4-dioxide derivatives.

Compounds **26c** and **64a**,**d** are also able to induce apoptosis in the myelogenous leukemia cell line and its doxorubicin-resistant subline K562/4 cell with the expression of P-glycoprotein (P-gp) [61,114].

## 5. Targeting Signaling Pathways in Cells by Quinoxaline 1,4-Dioxide Derivatives

The search for ways to selectively induce tumor cell death is one of the key goals in the development of new anticancer drugs. Therefore, the discovery of the hypoxia-selective cytotoxic mode of action of quinoxaline 1,4-dioxides stimulated further study of the mechanisms of their antitumor activity. It has been shown that, similarly to the antibacterial action, the antitumor potency of quinoxaline 1,4-dioxides is associated with their bioreduction in tumor cells by reductases expressed under hypoxic conditions, leading to the generation of ROS and the damage of DNA in malignant cells [166,167,168].

As noted above, quinoxaline 1,4-dioxide derivatives are able to trigger a cascade of various proapoptotic processes and suppress the mechanisms of cell resistance to xenobiotics [61,114,165]. It has been shown that quinoxaline derivatives induce tumor cell death via apoptosis [169,170,171]. Additionally, due to the outstanding hypoxic selectivity of several quinoxaline 1,4-dioxides, this class has a high potential for the development of drugs for the targeted treatment of solid tumors. Thus, previous works [61,114,165] have shown that the hypoxic selectivity of compounds **26c**, **64a**,**d**, and **99b** (Figure 12, Figure 15 and Figure 16) is associated with the inhibition of expression and activity of the hypoxia-induced factor HIF-1α in tumor cells. At the same time, amino derivatives **26c** and **64a**,**d** have a significant effect on the signaling pathways of the estrogen receptor ERα, which supports the growth of hormone-dependent tumor cells MCF-7 and can be considered as double blockers of HIF-1α/ERα, modulating the activity of the HIF-1α and ERK1/2 signaling pathways. In addition, the induction of apoptosis in HCT-116p53KO tumor cells with a deletion of the tumor suppressor p53, observed for derivative **63a** [60,61], is evidence that tumor cell death induced by quinoxaline 1,4-dioxides can proceed via a p53-independent mechanism [165,171].

The compounds **100** (Figure 16) and **99b** (Figure 15) could reduce the expression of vascular endothelial growth factor (VEGF) under hypoxic conditions. Quincetone (**104**, Figure 17) and derivative **100** (Figure 16) also down-regulated the expression of mitogen-activated protein kinases (MAPKs) involved in the mitochondrial apoptosis pathway. When studying the molecular mechanisms of action of olaquindox, D. Li and co-workers found that it can induce DNA damage and S-phase arrest, contributing to an increase in the expression of GADD45a, cyclin A, Cdk2, p21, and p53, a decrease in cyclin D1, as well as activation of phosphorylation of c-Jun *N*-terminal kinases (p-JNK), p38 (p-p38), and extracellular signal-regulated kinases (p-ERK) [172].

The A. Monge group found that the leishmanicidal activity of compound **92** (Figure 10) is associated with the induction of necrosis of *L. mexicana* caused by ROS production, leading to damage to the cell membrane, phosphatidylserine flip-flop, disruption of the mitochondrial membrane potential, and subsequent fragmentation of the parasite DNA [173]. The study of the mechanism of antiparasitic action showed that thioredoxin reductase (EhTrxR) is the main biological target of quinoxaline 1,4-dioxides in the cells of *E. histolytica* [174]. Compound **95** (Figure 11) acts as a substrate that disrupts the electron transfer in the active site of the EhTrxR enzyme, which leads to an increase in ROS levels and induces oxidative stress, causing changes in chromatin, cell granularity, and the redistribution of vacuoles.

## 6. Agricultural Use of Quinoxaline 1,4-Dioxide Derivatives

Quinoxaline 1,4-dioxide derivatives have been used for over 50 years as animal growth promoters, significantly improving feed efficiency in animal husbandry [175]. The most well-known compounds (**101**–**104**) are shown in Figure 17 [176,177].

Carbadox (CARB, **101**, Figure 17) acts on Gram-positive bacteria but is inactive against Gram-negative bacteria. It was patented by Pfizer in 1968. As a feed additive, it stimulates the growth of pigs and is also used to control swine dysentery (caused by *Serpulina hyodysenteriae*), bacterial enteritis (caused by *Salmonella* spp.), and nasal infections (caused by *Bordetella bronchiseptica*) [178]. Carbadox improves the retention of dietary proteins [23,24] and increases the ratio of RNA to DNA and protein to DNA in the lean muscles of young pigs [179]. Additionally, it reduces the energy requirement of the gastrointestinal tract, spleen, and pancreas in pigs [180]. However, the use of carbadox as a feed additive for livestock has been banned in Canada and the EU since 2004 due to the experimental evidence showing its carcinogenic and genotoxic properties [181]. Nonetheless, it is still used as a therapeutic and preventive agent in the treatment of bacterial enteritis of pigs and porcine dysentery caused by *B. hyodysenteriae* in the United States, Canada, and some other countries.

Olaquindox (OLAQ, **102**, Figure 17), developed by Bayer in 1967, is used to stimulate the growth of pigs. The drug improves the functioning of the gastrointestinal tract in animals, suppressing pathogenic Gram-positive and Gram-negative microflora. It is also used to prevent swine dysentery caused by *B. hyodysenteriae* [182]. Carbadox (**101**) and olaquindox (**102**) can irritate mucous membranes and cause superficial dermatitis and photoallergic reactions in animals [183].

Studies have shown that quinoxaline 1,4-dioxide derivatives are not present in meat after 10 days of keeping pigs before slaughter. Furthermore, carbadox and olaquindox exhibit low acute toxicity [183]. The LD_50_ for administration of carbadox peros in rats is 850 mg/kg, and in mice, it is 2810 mg/kg. Olaquindox (**102**) is somewhat less toxic, with an oral LD_50_ for rabbits ranging from 1000 to 2000 mg/kg and about 1600 mg/kg for rats. The subcutaneous LD_50_ of olaquindox in rats is 1275 mg/kg. However, both carbadox and olaquindox have shown genotoxicity using in vitro and in vivo studies. Carbadox (**101**) induced chromosome damage in human lymphocytes, and olaquindox damaged DNA in Chinese hamster cell culture. Additionally, carbadox (**101**) and olaquindox (**102**) caused an increase in the number of micronuclear polychromatic erythrocytes in the bone marrow of rats, indicating their genotoxicity in animals. The metabolites of carbadox (**101**) and olaquindox (**102**) could potentially be carcinogenic, thus requiring careful monitoring of their content in the meat and animal fat.

Mequindox (MEQ, **21b**, Figure 17), created by Pfizer, exhibits high antibacterial activity against Gram-negative bacteria, particularly *Salmonella* spp., and is widely used in China as a feed additive for animals and as a veterinary drug for dysentery and white diarrhea in pigs [184].

Quincetone (QUIN, **103**, Figure 17), developed by the Lanzhou Institute of Animal Husbandry and Veterinary Drugs (Chinese Academy of Agricultural Sciences, Lanzhou, China), is active against *B. hyodysenteriae*, *Salmonella* spp., *E. coli*, and other Gram-negative bacilli. Quincetone (**103**) is approved as an animal growth promoter in China and has been used in swine, poultry, and aquaculture since 2003 [185].

Cyadox (CYAD, **104**, Figure 17), described by Chemapol Benelux, exhibits antibacterial activity against *Staphylococcus hyicus*, *Pasteurella multocida*, *E. coli* and also shows a beneficial effect as a growth promoter in broiler chickens and pigs. Cyadox (**104**) has low toxicity and high safety, making it a potential substituent for olaquindox (**102**) and carbadox (**101**) in animal husbandry [186].

Several quinoxaline 1,4-dioxide derivatives, such as compounds **105** and **106** (Figure 18), have shown herbicidal activity against dicotyledonous and cereal plants by disrupting the functioning of photosystem I [187]. Due to the presence of two *N*-oxide groups, compounds **105** and **106** compete with NADP^+^ in the active site of ferredoxin-NADP^+^ reductase, initiating numerous oxidative transformations in plant cells, leading to their death.

In conclusion, the synthetic availability and the ability to modulate desired pharmacological properties make quinoxaline 1,4-dioxide derivatives as a promising class of heterocyclic compounds for the development of new agrochemical agents.

## 7. Toxicological Properties of Quinoxaline 1,4-Dioxides

The ability of quinoxaline 1,4-dioxides to generate ROS in cells, thereby inducing DNA single- or double-stranded breaks, also underlies their mutagenic action. For example, T. Negishi and co-workers proved the mutagenicity of compounds **101**–**104** (Figure 17) and **3** (Figure 2), as well as some other quinoxaline 1,4-dioxides, on *Salmonella typhimurium* TA 98 and TA 100 strains [188]. It was shown that the mutagenicity of quinoxaline 1,4-dioxides is associated with the presence of two *N*-oxide groups since quinoxaline and quinoxaline 1-*N*-oxide did not exhibit mutagenic activity under the same conditions. In addition, using in vivo experiments (a micronuclear test in mice) additionally confirmed the clastogenicity of quinoxaline 1,4-dioxides [22]. Thus, olaquindox (**102**, Figure 17) induces chromosome aberrations when administered orally at a dose of 100 mg/kg, while carbadox (**101**, Figure 17) exhibits the same effect at a dose of 800 mg/kg [189]. Nevertheless, in contrast to the derivatives **101** and **102**, the agricultural agent cyadox (**104**, Figure 17) did not show a damaging effect on chromosomes, even at a dose of 1200 mg/kg. It was found that derivative **101** has a teratogenic effect and leads to intrauterine malformation of the fetus and death of the embryo using in vivo experiments on rats [190]. Teratogenic, embryotoxic, and mutagenic effects have been shown for the clinically used dioxidine, which limits its use [191].

The phototoxicity of some derivatives of quinoxaline 1,4-dioxide was observed in [192], indicating the ability of quinoxaline 1,4-dioxides used in agriculture and veterinary practice to cause photoallergic reactions in animals. It was found that the reactive oxaziridine intermediate **29** and its 2,3-disubstituted derivatives, as well as its dioxaziridine analogs (Figure 8) generated under UV irradiation, engage in the chemical interaction with proteins. In particular, irradiation of 3-phenylquinoxaline-2-carbonitrile 1,4-dioxide (**89b**, Figure 9) by the long-wavelength part of the visible spectrum leads to selective cleavage of the bond at the 2’-position of deoxyguanosine [193]. An enhancement of photoinduced bond breaking in pBR322 plasmid DNA caused by compound **89b** (Figure 9) occurs with the further participation of NADPH-dependent enzymes under anaerobic conditions. Notably, in contrast to derivative **89b** (Figure 9), DNA damage induced by quinoxaline 1,4-dioxide (**28a**, Figure 8) was independent of NADPH-dependent enzymes.

## 8. Conclusions

The search for new possibilities to functionalize quinoxaline 1,4-dioxides has attracted special attention from chemists due to the unique biological properties and high pharmacological activity of this class of compounds. Over the past decade, several effective methods for their synthesis have been developed, and classical methods for the preparation and modification of quinoxaline 1,4-dioxides have been improved. Progress in the chemistry of this class has allowed the evaluation of bioactivities for a wide range of functionalized derivatives. Recent experimental works demonstrate the continued scientific and practical interest in quinoxaline 1,4-dioxides. Therefore, a systematization of preparation methods of quinoxaline 1,4-dioxides are in demand to develop new schemes for the synthesis of these compounds and contribute to further evaluations of this class of heterocycles.

In recent years, there has been a high priority placed on modifying the structure of quinoxaline 1,4-dioxide to develop drugs with greater selectivity or improved therapeutic properties. However, analyzing the relationship between biological activity and the structure of quinoxaline 1,4-dioxides, considering multiple types of activity simultaneously, and evaluating specific structural features, is a challenging task. General observations include the following:
-the introduction of electron-withdrawing substituents at positions 3 and 6/7 of the quinoxaline moiety promotes more readily reduction by bacterial or eukaryotic reductases, resulting in increased cytotoxicity and antimicrobial activity of such derivatives;-the presence of a carbonitrile moiety at position 2 of the pyrazine ring enhances antiproliferative activity, while ester or carboxamide fragments at this position improve the antibacterial properties of quinoxaline 1,4-dioxides;-overall, the presence of a halogen atom in the quinoxaline core enhances the biological activity of the compounds, with chlorine being the most significant;-the introduction of electron-donating groups at C3 or C7 carbon atoms of the heterocycle increases the hypoxic cytotoxicity of quinoxaline-2-carbonitrile 1,4-dioxide derivatives.

However, the activity of such compounds strongly depends on the structure of the substituents. The presence of linear alkylamino groups at these positions had a negative effect, resulting in a complete loss of antiproliferative properties and selectivity under hypoxic conditions. [61,108]. On the other hand, the presence of cyclic amino groups at positions 3 or 7 leads to a significant increase in the hypoxic selectivity of quinoxaline-2-carbonitrile 1,4-dioxides. Additionally:
-lipophilic substituents at positions 2, 6, or 7 improve the antituberculous potential of the derivatives;-acylamino groups at position 3 of the quinoxaline ring have a positive effect on suppressing the growth of protozoa (*Leishmania* spp. and *Plasmodium* spp);-the introduction of salt-forming fragments leads to water-soluble derivatives while retaining their biological potency.


However, due to the limited available data on target proteins and the necessity for further searches for intracellular reductases involved in the activation of quinoxaline 1,4-dioxides, a comprehensive understanding of the mechanism of action remains essential. H. Zhang et al. conducted QSAR analysis for antimycobacterial activity using CoMFA and CoMSIA models [89]. The results of 3D-QSAR modeling identified several critical requirements for the structure of quinoxaline 1,4-dioxides for their antituberculosis properties. This indicates the specificity of the binding of this class of compounds to intracellular targets important for inhibiting pathogen growth. Electron-withdrawing and sterically bulky substituents, particularly halogen atoms, at position 7 of the quinoxaline are preferred for increasing antimycobacterial activity. Furthermore, the introduction of hydrogen bond donors at the C_2_ carbon atom of the pyrazine ring enhances activity against *Mycobacterium* spp.

Despite the observed progress, the further development of methods for preparing new quinoxaline 1,4-dioxides and the study of their mode of action remains an urgent task. Addressing these tasks could broaden the practical applications for this unique class of heterocyclic compounds.

## Data Availability

Data sharing not applicable.

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
