# Peer review of "Quinoxaline 1,4-Dioxides: Advances in Chemistry and Chemotherapeutic Drug Development"

_pharmaceuticals, 2023, doi:10.3390/ph16081174_

Round 1
Reviewer 1 Report
In this manuscript G. I. Buravchenko and A. E. Shchekotikhin „ Quinoxaline 1,4-dioxides: Advances in chemistry and chemo therapeutic drug development“ describe a preparation of the library of heterocyclic N-oxides, their biological and pharmaceutical activity and therapeutic application. Overall, the manuscript would be of interest to readers as it makes a good addition to the chemical and applied knowledge of the field. However, the manuscript must be very carefully checked and corrected.
The article is written in fairly good English, but the sentence structure should still be observed, which is irregular. For example "Abstract: Due to the diverse biological properties of N-oxides of heterocyclic compounds, they are in demand in various fields of medical chemistry." Also "Since the second half of the twentieth century, the high antibacterial activity of synthetic quinoxaline 1,4-dioxides have been observed.", and elsewhere.The manuscript needs minor revision and correction of the remaining inaccuracies:
L45: (2) – Bold.
L101, 151, 167, 202, 203, 208, 214, 218, 226, 265, 275, 278, 359, 475, 539: N – Italic.
L113: o- - Italic.
L148, 166, 296: β- Italic.
L236: di-O-acetyl… -O- Italic.
L287: p-… -Italic.
L347: N,N,S - Italic.
L395, 408, 408: ..amino derivatives…
It is necessary to standardize the writing of concentrations in the descriptions of biological studies. Either µg/ml or µg/mL.
References list should be checked and corrected according the Guidelines.
The article is written in fairly good English, but the sentence structure should still be observed, which is irregular. For example "Abstract: Due to the diverse biological properties of N-oxides of heterocyclic compounds, they are in demand in various fields of medical chemistry." Also "Since the second half of the twentieth century, the high antibacterial activity of synthetic quinoxaline 1,4-dioxides have been observed.", and elsewhere.Author Response
Dear colleague,
We are thankful careful consideration of our manuscript and for valuable criticism. Following of your comments and suggestions from other reviewers, we have edited a text and added some corrections in the manuscript. We submit the revised manuscript with all changes highlighted in blue.
Thank you again for your consideration.
Sincerely,
Andrey E. Shchekotikhin and Galina I. Buravchenko
- The article is written in fairly good English, but the sentence structure should still be observed, which is irregular. For example "Abstract: Due to the diverse biological properties of N-oxides of heterocyclic compounds, they are in demand in various fields of medical chemistry." Also "Since the second half of the twentieth century, the high antibacterial activity of synthetic quinoxaline 1,4-dioxides have been observed.", and elsewhere.
Reply: Thank you for your advice. We have corrected the text.
- L45: (2) – Bold.
L101, 151, 167, 202, 203, 208, 214, 218, 226, 265, 275, 278, 359, 475, 539: N – Italic.
L113: o- - Italic.
L148, 166, 296: β- Italic.
L236: di-O-acetyl… -O- Italic.
L287: p-… -Italic.
L347: N,N,S - Italic.
L395, 408, 408: ..amino derivatives….
Reply: We apology for these mistakes: in the initial file, all letter designations were indicated in italics, but after uploading the file, a text formatting was failed. All specified lines was revised again and corrected. Please check the pdf-version of the file.
- It is necessary to standardize the writing of concentrations in the descriptions of biological studies. Either µg/ml or µg/mL.
Reply: Thank you for your suggestion. The concentration values in the text were thoroughly revised and updated.
- References list should be checked and corrected according the Guidelines.
Reply: Thank you for your suggestion. All references were thoroughly revised and corrected.

Reviewer 2 Report
The submitted manuscript is an interesting review paper.
The review applies the selected methods for the synthesis and directions in the chemical modification of quinoxaline 1,4-dioxide derivatives and biological properties. Presents the prospects for the practical application of the seleted compounds.
However, the manuscript contains some mistakes, this requires correction and amendments.
For example
1) page 23, Fig 12: In all compounds 98, the substituent R1 is H, so there is no need to indicate it in the chemical formula.
2) The abbreviations of the compounds used should be explained.
For example - DCQ (6,7-dichloroquinoxaline ?).
They do not need to be used if the authors use its number in the text DCQ (99b) - Fig. 13, line 734
Author Response
Dear colleague,
We are thankful careful consideration of our manuscript and for valuable criticism. Following of your comments and suggestions from other reviewers, we have edited a text and added some corrections in the manuscript. We submit the revised manuscript with all changes highlighted in blue.
Thank you again for your consideration.
Sincerely,
Andrey E. Shchekotikhin and Galina I. Buravchenko
- Page 23, Fig 12: In all compounds 98, the substituent R1 is H, so there is no need to indicate it in the chemical formula.
Reply: Thank you for your suggestion. The structure of compounds 98 was revised.
- The abbreviations of the compounds used should be explained.
For example: - DCQ (6,7-dichloroquinoxaline ?). They do not need to be used if the authors use its number in the text DCQ (99b) - Fig. 13, line 734.
Reply: Thank you for your suggestion. We removed DCQ abbreviations because it is not commonly accepted.

Reviewer 3 Report
Query#1
In the introduction I kindly ask to clearly introduce the importance of nitrogen heterocycles, as key pharmacophoric moieties, for the synthesis of novel anticancer compounds. Indeed, the presence of different nitrogen electron-donor atoms in the structure improves the interaction with target proteins, enzymes, and receptors through the formation of several types of interaction, such as: hydrogen bonds, dipole-dipole, hydrophobic interactions, van der Waals forces and π-stacking interactions.
At this purpose I suggest to the authors to cite the following paper:
-Vitaku, E., Smith, D. T., & Njardarson, J. T. (2014). Analysis of the structural diversity, substitution patterns, and frequency of nitrogen heterocycles among U.S. FDA approved pharmaceuticals. Journal of medicinal chemistry, 57(24), 10257–10274. https://doi.org/10.1021/jm501100b.
- Mohan, C. D., Anilkumar, N. C., Rangappa, S., Shanmugam, M. K., Mishra, S., Chinnathambi, A., Alharbi, S. A., Bhattacharjee, A., Sethi, G., Kumar, A. P., Basappa, & Rangappa, K. S. (2018). Novel 1,3,4-Oxadiazole Induces Anticancer Activity by Targeting NF-κB in Hepatocellular Carcinoma Cells. Frontiers in oncology, 8, 42. https://doi.org/10.3389/fonc.2018.00042
- Pecoraro, C., De Franco, M., Carbone, D., Bassani, D., Pavan, M., Cascioferro, S., Parrino, B., Cirrincione, G., Dall'Acqua, S., Moro, S., Gandin, V., & Diana, P. (2023). 1,2,4-Amino-triazine derivatives as pyruvate dehydrogenase kinase inhibitors: Synthesis and pharmacological evaluation. European journal of medicinal chemistry, 249, 115134. https://doi.org/10.1016/j.ejmech.2023.115134
-Moniot, S., Forgione, M., Lucidi, A., Hailu, G. S., Nebbioso, A., Carafa, V., Baratta, F., Altucci, L., Giacché, N., Passeri, D., Pellicciari, R., Mai, A., Steegborn, C., & Rotili, D. (2017). Development of 1,2,4-Oxadiazoles as Potent and Selective Inhibitors of the Human Deacetylase Sirtuin 2: Structure-Activity Relationship, X-ray Crystal Structure, and Anticancer Activity. Journal of medicinal chemistry, 60(6), 2344–2360. https://doi.org/10.1021/acs.jmedchem.6b01609
-Carbone, D., De Franco, M., Pecoraro, C., Bassani, D., Pavan, M., Cascioferro, S., Parrino, B., Cirrincione, G., Dall'Acqua, S., Moro, S., Gandin, V., & Diana, P. (2023). Discovery of the 3-Amino-1,2,4-triazine-Based Library as Selective PDK1 Inhibitors with Therapeutic Potential in Highly Aggressive Pancreatic Ductal Adenocarcinoma. International journal of molecular sciences, 24(4), 3679. https://doi.org/10.3390/ijms24043679
- Abdel-Aziz, A. K., Dokla, E. M. E., Abouzid, K. A. M., & Minucci, S. (2022). Discovery of EMD37, a 1,2,4-oxadiazole derivative, as a novel endoplasmic reticulum stress inducer with potent anticancer activity. Biochemical pharmacology, 206, 115316. https://doi.org/10.1016/j.bcp.2022.115316
Query#2
In the paragraph introduction, it is clear the main objective of the present review, however, I suggest to the authors to remove the part in which they report different examples of other reviews on quinoxaline N-oxides, provided by other authors. It is sufficient to say that most of the reviews published in the last twenty years have focused on biological aspects, but limited analysis of the structure-activity relationships has been reported.
Query#3
Section 2. Methods of the synthesis of quinoxaline 1,4-dioxides, I suggest to the authors to highlight the synthetic route taken for obtaining quinoxaline 1,4-dioxiedes, utilizing a bullet point or a table, (e.g., by oxidation, by cyclization of o-benzoquinone with 1,2-dicarbonyl compounds, by cyclization of benzofuroxanes with enamines...)
Query#4
Section 3.2 Reactions of aminoderivatives of quinoxaline 1,4-dioxides, the authors provide a fascinating overview of the reaction of the aminoquinoxaline with acid chlorides, further the ability of aminoquinoxalines to develop metal-containing chelates was reported, however I suggest to the authors to organize this section better.
When referring to a bacterium species, the name should be in italics, please revise the entire article.
Please, report “in vitro”, “in vivo” in italics.
The authors are advised to thoroughly revise the style and form of the English language used in the manuscript.
Author Response
Dear colleague,
We are thankful careful consideration of our manuscript and for valuable criticism. Following of your comments and suggestions from other reviewers, we have edited a text and added some corrections in the manuscript. We submit the revised manuscript with all changes highlighted in blue.
Thank you again for your consideration.
Sincerely,
Andrey E. Shchekotikhin and Galina I. Buravchenko
- In the introduction I kindly ask to clearly introduce the importance of nitrogen heterocycles, as key pharmacophoric moieties, for the synthesis of novel anticancer compounds. Indeed, the presence of different nitrogen electron-donor atoms in the structure improves the interaction with target proteins, enzymes, and receptors through the formation of several types of interaction, such as: hydrogen bonds, dipole-dipole, hydrophobic interactions, van der Waals forces and π-stacking interactions.
At this purpose I suggest to the authors to cite the following paper:
-Vitaku, E., Smith, D. T., & Njardarson, J. T. (2014). Analysis of the structural diversity, substitution patterns, and frequency of nitrogen heterocycles among U.S. FDA approved pharmaceuticals. Journal of medicinal chemistry, 57(24), 10257–10274. https://doi.org/10.1021/jm501100b.
- Mohan, C. D., Anilkumar, N. C., Rangappa, S., Shanmugam, M. K., Mishra, S., Chinnathambi, A., Alharbi, S. A., Bhattacharjee, A., Sethi, G., Kumar, A. P., Basappa, & Rangappa, K. S. (2018). Novel 1,3,4-Oxadiazole Induces Anticancer Activity by Targeting NF-κB in Hepatocellular Carcinoma Cells. Frontiers in oncology, 8, 42. https://doi.org/10.3389/fonc.2018.00042
- Pecoraro, C., De Franco, M., Carbone, D., Bassani, D., Pavan, M., Cascioferro, S., Parrino, B., Cirrincione, G., Dall'Acqua, S., Moro, S., Gandin, V., & Diana, P. (2023). 1,2,4-Amino-triazine derivatives as pyruvate dehydrogenase kinase inhibitors: Synthesis and pharmacological evaluation. European journal of medicinal chemistry, 249, 115134. https://doi.org/10.1016/j.ejmech.2023.115134
-Moniot, S., Forgione, M., Lucidi, A., Hailu, G. S., Nebbioso, A., Carafa, V., Baratta, F., Altucci, L., Giacché, N., Passeri, D., Pellicciari, R., Mai, A., Steegborn, C., & Rotili, D. (2017). Development of 1,2,4-Oxadiazoles as Potent and Selective Inhibitors of the Human Deacetylase Sirtuin 2: Structure-Activity Relationship, X-ray Crystal Structure, and Anticancer Activity. Journal of medicinal chemistry, 60(6), 2344–2360. https://doi.org/10.1021/acs.jmedchem.6b01609
-Carbone, D., De Franco, M., Pecoraro, C., Bassani, D., Pavan, M., Cascioferro, S., Parrino, B., Cirrincione, G., Dall'Acqua, S., Moro, S., Gandin, V., & Diana, P. (2023). Discovery of the 3-Amino-1,2,4-triazine-Based Library as Selective PDK1 Inhibitors with Therapeutic Potential in Highly Aggressive Pancreatic Ductal Adenocarcinoma. International journal of molecular sciences, 24(4), 3679. https://doi.org/10.3390/ijms24043679
- Abdel-Aziz, A. K., Dokla, E. M. E., Abouzid, K. A. M., & Minucci, S. (2022). Discovery of EMD37, a 1,2,4-oxadiazole derivative, as a novel endoplasmic reticulum stress inducer with potent anticancer activity. Biochemical pharmacology, 206, 115316. https://doi.org/10.1016/j.bcp.2022.115316.
Reply: Thank you for your valuable suggestion. We revised the Introduction of the review and add corresponding references.
- In the paragraph introduction, it is clear the main objective of the present review, however, I suggest to the authors to remove the part in which they report different examples of other reviews on quinoxaline N-oxides, provided by other authors. It is sufficient to say that most of the reviews published in the last twenty years have focused on biological aspects, but limited analysis of the structure-activity relationships has been reported.
Reply: Thank you for your suggestion. The text was revised in accordance of your recommendations.
- Section 2. Methods of the synthesis of quinoxaline 1,4-dioxides, I suggest to the authors to highlight the synthetic route taken for obtaining quinoxaline 1,4-dioxiedes, utilizing a bullet point or a table, (e.g., by oxidation, by cyclization of o-benzoquinone with 1,2-dicarbonyl compounds, by cyclization of benzofuroxanes with enamines...).
Reply: Thank you for your valuable suggestion. We resumed the synthetic approaches as bullet points and schemes on figure 4.
- The Section 3.2. Reactions of aminoderivatives of quinoxaline 1,4-dioxides, the authors provide a fascinating overview of the reaction of the aminoquinoxaline with acid chlorides, further the ability of aminoquinoxalines to develop metal-containing chelates was reported, however I suggest to the authors to organize this section better.
Reply: Thank you for your suggestion. The text was revised in accordance of your recommendations.
- When referring to a bacterium species, the name should be in italics, please revise the entire article.
- Please, report “in vitro”, “in vivo” in italics.
Reply: We apology for these mistakes: in the initial file, all letter designations were indicated in italics, but after uploading the file, a text formatting was failed. All specified lines was revised again and corrected. Please check the pdf-version of the file.
- The authors are advised to thoroughly revise the style and form of the English language used in the manuscript.
Reply: The remark was taken into account and the text was carefully revised.

Reviewer 4 Report
In this manuscript, Buravchenko and Shchekotikhin describes the chemistry and medicinal importance of quinoxaline 1,4-dioxides. Authors have covered a significant literature on the interesting topic. I recommend it for publication if authors address the following comments.
1. The medicinal section lacks a descriptive SAR. A very brief summary is presented. Authors are encouraged to add more details.
2. In synthetic chemistry section, no mechanistic details are given. Authors should add mechanisms wherever necessary.
3. Delete full stop and semi-colon in all the schemes.
4. Bacterial strains etc should be italicized throughout the manuscript.
5. Authors should add legends to all the schemes.
6. In all schemes, there should be space between temperature digits and units and same for reaction time.
7. Authors should be consistent in providing the reaction conditions. Some missing time, temperature etc. check all schemes.
Minor editing required.
Author Response
Dear colleague,
We are thankful careful consideration of our manuscript and for valuable criticism. Following of your comments and suggestions from other reviewers, we have edited a text and added some corrections in the manuscript. We submit the revised manuscript with all changes highlighted in blue.
Thank you again for your consideration.
Sincerely,
Andrey E. Shchekotikhin and Galina I. Buravchenko
- The medicinal section lacks a descriptive SAR. A very brief summary is presented. Authors are encouraged to add more details.
Reply: Thank you for your valuable suggestion. In accordance of your recommendations, the main featurettes peculiarities of SAR for different kinds of biological activities were added.
- In synthetic chemistry section, no mechanistic details are given. Authors should add mechanisms wherever necessary.
Reply: Thank you for your valuable suggestion. In accordance of your recommendations, a mechanism of the Beirut reaction was added
- Delete full stop and semi-colon in all the schemes.
Reply: Thank you for your suggestion. The schemes were revised in accordance of your recommendations.
- Bacterial strains etc should be italicized throughout the manuscript.
Reply: We apology for these mistakes: in the initial file, all letter designations were indicated in italics, but after uploading the file, a text formatting was failed. All specified lines was revised again and corrected. Please check the pdf-version of the file.
- Authors should add legends to all the schemes.
Reply: Thank you for your valuable suggestion. The legends for all schemes were added in accordance of your recommendations.
- In all schemes, there should be space between temperature digits and units and same for reaction time.
Reply: Thank you for your suggestion. The schemes were revised in accordance of your recommendations.
- Authors should be consistent in providing the reaction conditions. Some missing time, temperature etc. check all schemes.
Reply: Thank you for your suggestion. The schemes were revised in accordance of your recommendations.

Round 2
Reviewer 4 Report
Authors have improved the manuscript. It could be accepted now.
minor inconsistencies